# Methamphetamine Induces Metallothionein 1 Expression and an Inflammatory Phenotype in Primary Human HIV-Infected Macrophages

**DOI:** 10.3390/ijms26188875

**Published:** 2025-09-12

**Authors:** Jessica Weiselberg, Meng Niu, Cristian A. Hernandez, Howard S. Fox, Tina M. Calderon, Joan W. Berman

**Affiliations:** 1Department of Pathology, Albert Einstein College of Medicine, 1300 Morris Park Ave, Bronx, NY 10461, USA; jessica.weiselberg@einsteinmed.edu (J.W.); cristian.hernandez@einsteinmed.edu (C.A.H.); tina.calderon@einsteinmed.edu (T.M.C.); 2Department of Neurological Sciences, University of Nebraska Medical Center, Omaha, NE 68198, USA; meng.niu@unmc.edu (M.N.); hfox@unmc.edu (H.S.F.)

**Keywords:** HIV-NCI, methamphetamine, myeloid cells, metallothionein 1, cytokines, reactive oxygen species

## Abstract

HIV-associated neurocognitive impairment (HIV-NCI), a comorbidity of human immunodeficiency virus (HIV) infection, affects up to 50% of people with HIV (PWH). HIV-infected monocytes that transmigrate across the blood–brain barrier and mature into macrophages establish a central nervous system (CNS) viral reservoir that activates and infects parenchymal cells, contributing to neuronal damage that characterizes HIV-NCI. Methamphetamine (meth) use is prevalent in PWH and further impairs cognitive functioning. To examine whether meth-mediated dysregulation of macrophage functions may contribute to increased HIV-NCI, we characterized differential gene expression in primary human HIV-infected macrophages treated daily with meth for five days by RNA-sequencing. We identified increases in multiple gene isoforms of metallothionein 1 (MT1), a heavy metal binding protein involved in protective mechanisms against metal toxicity and oxidative stress. Nuclear localization of MT1 protein was previously shown to either positively or negatively affect nuclear factor κB (NF-κB) activity in a cell type specific manner, with nuclear MT1 contributing to LPS-induced TNF-α and IL-6 in macrophages. We found that daily meth treatment for one to five days increased nuclear localization of MT1 in macrophages acutely infected with HIV which was associated with increased LPS-induced CXCL8 and CCL8, and a trend towards increased basal and/or LPS-induced expression of other cytokines/chemokines, including TNF-α and IL-6, that was donor specific. Reactive oxygen species (ROS) levels were not changed with meth treatment although there was a donor specific trend towards increased ROS with multiple days of meth treatment. These data indicate that repeated exposure of macrophages to meth in the context of HIV increases nuclear MT1 localization, which is associated with increased inflammatory mediator production, and therefore may be a mechanism that contributes to meth-mediated exacerbation of HIV-NCI.

## 1. Introduction

HIV-associated neurocognitive impairment (HIV-NCI) impacts up to 50% of people with HIV (PWH), despite viral suppression with antiretroviral therapy (ART) [1,2,3,4]. HIV-NCI manifests as a spectrum of cognitive and motor deficits that significantly impair quality of life [5,6] and is an independent risk factor for mortality in PWH [7,8]. It is mediated, in part, by peripheral HIV

Infected monocytes transmigrate across the blood–brain barrier (BBB) and differentiate into long-lived infected macrophages [9,10], establishing a viral reservoir in the central nervous system (CNS), even with suppressive ART [11,12,13,14,15,16]. These reservoirs promote HIV infection and/or activation of CNS cells including perivascular and parenchymal macrophages, microglia, and astrocytes. This results in chronic production of cytokines and chemokines, reactive oxygen species (ROS), and other host and viral factors that promote a neuroinflammatory and neurotoxic environment, mediating neuronal damage that accumulates over time and manifests as HIV-NCI [10,17,18,19]. ART mitigates CNS inflammation, but it does not reduce it to levels comparable to a person without HIV [20,21]. Thus, HIV-NCI persists in PWH on suppressive ART. Additionally, viral reservoirs in the CNS are difficult to target, which makes developing therapies for reducing HIV-NCI a challenge [22].

Substance use is a growing public health crisis that intersects with the HIV epidemic. Methamphetamine (meth) is a potent CNS stimulant that causes neuronal excitotoxicity and damage that can result in cognitive impairment [23,24,25]. Meth use has increased over time, with 2.6 million people in the United States alone using meth in 2023 [26]. There appears to be an increased prevalence of meth use among PWH compared to the general population, as the number of people who use meth in the U.S. is less than one percent while two recent cohort studies showed that the prevalence of meth use among PWH was 17% [27] and 6.88% [28], respectively. Studies demonstrated that some PWH who use meth have worse cognitive outcomes than PWH who do not use meth [29,30,31]. The mechanisms by which meth increases cognitive dysfunctions in PWH are not completely characterized. Meth use in PWH is linked to increased viral loads due to less stringent adherence to ART regimens [32,33]. Increased viral loads in PWH who use meth also occurs despite ART adherence [34,35]. Additionally, PWH who use meth have increased T-cell activation, proliferation, and exhaustion, as well as more HIV DNA copies in peripheral blood mononuclear cells (PBMC) compared to PWH who do not use meth [36]. These findings suggest that meth may be contributing to the chronic inflammation characteristic of HIV neuropathogenesis through immune activation mechanisms, even in PWH on ART, and highlights the need to examine further the impact of meth use on HIV-NCI.

Macrophages are an important innate immune cell that perform a variety of functions to maintain cellular and tissue homeostasis. These functions, including cytokine and ROS production, antigen presentation, and phagocytosis, must be regulated to inhibit inflammatory responses contributing to tissue damage and disease pathogenesis, particularly in the CNS [37,38]. Dysregulation of macrophage functions upon their infection with HIV likely contributes to HIV-NCI pathogenesis [39,40,41]. Macrophage functions may be dysregulated further by meth, diminishing their ability to maintain CNS homeostasis, resulting in increased HIV-NCI pathogenesis. Our laboratory has focused on meth-mediated effects on macrophages, and we previously reported that HIV-infected macrophages treated once with meth for 24 h exhibited decreased phagocytosis and autophagy, and increased ROS production [42].

In individuals, including PWH, meth is most often used chronically over extended time periods. Therefore, in this study we modeled the impact of this type of meth use during acute HIV infection by treating HIV-infected primary human macrophages daily with meth for five days and then analyzed RNA isolated from these cells by bulk RNA-sequencing (RNA-seq). Of the differentially expressed genes (DEG) induced by repeated meth treatment, we focused on metallothionein 1 (*MT1*) genes. MT1 is a small heavy metal binding protein that performs many functions including ROS and heavy metal neutralization, protection against apoptosis, and cell type specific modulation of cytokine production [43,44,45,46]. When localized to the cytoplasm, MT1 can neutralize ROS to alleviate oxidative stress. However, during cell stress, proliferation, and differentiation, it can localize to the nucleus where it contributes to regulation of transcriptional pathways, including oxidative stress responses. MT1 in the cytoplasm or nucleus can either activate or inhibit NF-κB and cytokine production in a cell type specific manner, with MT1 exhibiting either inflammatory or anti-inflammatory properties [47,48,49,50,51,52,53]. In macrophages, MT1 contributes to LPS-induced TNF-α and IL-6 by a proposed mechanism involving interactions between nuclear MT1 and NF-κB [44]. These data suggest that nuclear localization of MT1 contributes to an increase in the inflammatory phenotype of macrophages. There have been studies demonstrating increased MT1 expression in neurodegenerative conditions such as Parkinson’s and Alzheimer’s Diseases, where it is proposed to be neuroprotective [54,55,56], but none examining the contributions of MT1 expression in CNS cells to the pathogenesis of HIV-NCI. Thus, this study used macrophages acutely infected with HIV to determine whether repeated meth treatment that models chronic use increases nuclear MT1 expression. In addition, we characterized basal and LPS-induced production of macrophage-related cytokines/chemokines and inflammatory mediators, including TNF-α, IL-6, and ROS after meth treatment to examine whether increased MT1 nuclear localization is associated with dysregulated macrophage inflammatory functions.

To our knowledge, our study is the first to demonstrate increased MT1 in the context of HIV and substance use. We confirmed RNA-seq data of increased MT1 gene expression by qRT-PCR showed increased nuclear localization of MT1 protein, and demonstrated increased LPS-induced cytokine/chemokine production in HIV-infected macrophages after meth treatment. Our findings indicate that upregulation and nuclear localization of MT1 in HIV-infected macrophages with repeated meth treatment is associated with increased cytokine/chemokine production. These data suggest that this may be a mechanism by which chronic meth use increases the inflammatory phenotype of macrophages in the CNS, contributing to increased HIV-NCI.

## 2. Results

### 2.1. Methamphetamine Does Not Change HIV Infection Levels and Is Not Toxic to HIV-Infected Macrophages

To model the effects of chronic meth use on CNS macrophage functions in PWH, we performed five days of repeated daily meth treatment on acutely infected primary human monocyte-derived macrophages (MDM). MDM were derived either from PBMC or from negatively or positively isolated monocytes from PBMC after six days of culture with macrophage colony stimulating factor (M-CSF). Cells were then infected with 20 ng/mL HIV_ADA_ and on the third day post-infection, treated daily with 50 μM meth or with media as an untreated control for one to five days.

Some studies showed that meth treatment increases HIV infection of macrophages when cells are treated both pre- and post-infection [57,58]. To address whether our meth treatment protocol increases HIV infection of macrophages, we collected supernatants from HIV-infected macrophages treated with meth or untreated and measured p24, the HIV capsid protein, which indicates productive infection. We did not find changes in p24 in supernatants with meth treatment at either the end of the five days of treatment (Figure 1A) or at each day of the five-day time course (Figure 1B). Thus, meth treatment starting at three days post-infection through eight days post infection does not change infection levels at any timepoint.

We also assessed whether meth is toxic to HIV-infected macrophages. Supernatants were collected after each day of treatment and LDH release was measured by cytotoxicity assay. We did not detect differences in LDH release with meth treatment compared to untreated at any timepoint, indicating that daily meth treatment is not toxic at the concentration used at any timepoint (Figure 2). Taken together, the findings that meth does not change HIV p24 and is not toxic to cells indicate that any changes we find are not due to increased HIV infection levels or upregulation of cell death mechanisms at each day of treatment, but rather to effects of meth in the context of HIV infection.

### 2.2. RNA Sequencing of HIV-Infected Macrophages Identifies Increased Expression of MT1 Genes with Methamphetamine Treatment

Macrophages derived from negatively isolated monocytes (three independent donors) or total PBMC (two independent donors) were infected and then untreated or treated daily with 50 µM meth for five days as described previously. RNA was extracted and sent to Novogene for bulk RNA-seq analysis. For one donor, we extracted RNA from triplicate plates to assess reproducibility of the sequencing data. Analysis of RNA from the triplicate plates showed high similarity as indicated by Pearson correlation coefficients R^2^ = 0.98 or greater (Appendix A). These data demonstrate that the differences in gene expression detected are due to biological variability among macrophages from different donors and not due to technical differences in the sequencing.

DEG data were pooled from all five donors, analyzed, and grouped by macrophage functions. In general, we found increased expression of metallothionein 1 (MT1) genes *MT1X*, *MT1H*, *MT1F*, *MT1G*, and *MT1E* in meth treated compared to untreated HIV-infected macrophages (Table 1). Due to many gene duplication events at this locus, there are several gene isoforms of MT1, with ten functional genes and eight pseudogenes [59,60,61]. All the MT1 genes that were differentially expressed in our dataset are functional isoforms.

HIV-infected macrophages from five independent donors, three derived from negatively isolated monocytes and two from PBMC, were treated with meth or left untreated for five days. After five days of treatment, RNA was extracted and bulk RNA-seq was performed. Differentially expressed gene (DEG) data were analyzed and grouped by macrophage functions. Upregulation of metallothionein 1 (*MT1*) genes was identified. Fold change DEG values for individual donors of HIV-infected macrophages treated with meth compared to HIV-infected untreated macrophages are shown for MT1 gene isoforms. Bars represent mean fold change of all donors.

### 2.3. qRT-PCR Validation of MT1 Genes and Maximal Expression Analysis Demonstrates Significant Increases in MT1 Gene Expression in HIV-Infected Macrophages Treated with Methamphetamine

To validate the RNA-seq data for MT1 gene expression, macrophages derived from PBMC from additional independent donors, were infected and treated daily with meth or untreated for five days, as described previously. We then extracted RNA, synthesized cDNA, and performed qRT-PCR for the differentially expressed MT1 genes: *MT1X*, *MT1H*, *MT1G*, *MT1F*, and *MT1E*. Our results showed a significant increase in *MT1X* expression (Figure 3A), and trends towards increased expression of *MT1H* (Figure 3B), *MT1G* (Figure 3C), and *MT1F* (Figure 3D), and no significant increase in *MT1E* (Figure 3E) for nine independent donors after five days of daily meth treatment compared to untreated, with untreated set to one. There was considerable variability in gene expression among donors, as is seen with primary human cells. We also quantified the expression of MT1 genes by qRT-PCR in uninfected monocyte-derived macrophages that were treated daily with meth or untreated for five days. *MT1X*, *MT1H*, *MT1G*, and *MT1F* expression was not increased in uninfected meth treated cells compared to untreated (Appendix A). These data indicate that MT1 gene expression in macrophages is increased by repeated meth treatment in the context of HIV infection.

Given the inherent differences in primary cells derived from different donors, we examined how meth impacts MT1 gene expression after each day of the five days of treatment. We isolated RNA from macrophage cultures derived from PBMC of additional independent donors after each day of treatment, synthesized cDNA, and performed qRT-PCR. This protocol was designed to account for donor dependent kinetic differences in meth-mediated increases in MT1 gene expression. The data were represented as the maximal meth-mediated increase in expression of each MT1 gene that occurred on any one of the five days of treatment for each donor. We found significant maximal increases in *MT1X* (Figure 4A), *MT1H* (Figure 4B), *MT1G* (Figure 4C), *MT1F* (Figure 4D) and *MT1E* (Figure 4E). Expression of each MT1 gene after each day of meth treatment from each donor are shown in Appendix A.

These data indicate that meth increases expression of MT1 genes in HIV-infected macrophages. We demonstrated that our meth treatment protocol did not increase HIV infection levels (Figure 1). There was also no correlation between expression of MT1 genes in all donors at all treatment times with HIV p24 levels (Appendix A). These data further confirm that meth-induced expression of MT1 genes in macrophages is not a consequence of increased HIV infection levels, but rather an effect of meth in the context of HIV infection. Since maximal expression of MT1 genes occurred at different days after meth treatment in macrophages from different donors, all subsequent functional assays were performed as a time course of meth treatment. This enabled us to characterize associations between increased MT1 expression that occurred after different days of meth treatment with functional properties of HIV-infected macrophages.

### 2.4. Methamphetamine Treatment Increases Nuclear Localization of MT1 in HIV-Infected Macrophages

MT1 is located primarily in the cytoplasm and localizes to the nucleus in response to cell stress, proliferation, and differentiation [49,50,52,53]. This can result in upregulation of its own transcription as well as upregulation of other ROS response mechanisms, and additional pro-survival mechanisms including apoptosis inhibition [48,49,51,53]. To assess whether meth treatment affects MT1 localization, we performed immunofluorescence staining and confocal microscopy for MT1 protein in the nucleus. Monocytes isolated by positive selection from nine independent donors were plated on glass coverslips and differentiated into macrophages, infected, and treated daily with meth or untreated, as described previously. Cells were fixed and stained for MT1 after each day of treatment. We quantified nuclear localization of MT1 and detected differences with meth treatment as detected by confocal Z-series images taken at high magnification to visualize the nucleus. Images of cells from a representative donor, shown in Figure 5A, demonstrate increased MT1 in the nucleus with meth treatment as indicated by the nuclear puncta. An IgG isotype matched negative control antibody was used to account for background fluorescence for each condition and minimal to no fluorescence was detected. For every donor there were one or more timepoints at which there is increased nuclear MT1 with meth treatment compared to untreated. The fold change values of maximal increase for each donor were pooled, and we found a significant increase in nuclear localization of MT1 with meth treatment compared to untreated (Figure 5B). The quantification of MT1 fluorescent signal per nucleus for untreated and meth treated cells from each donor after each day of the five-day time course is shown in Appendix A. Fold changes in nuclear MT1 signal for each donor for each day of meth treatment compared to untreated is shown in Appendix A. Given the donor variability, we did not find one timepoint of meth treatment that significantly increased MT1nuclear localization in cells from all donors, except that there was a trend towards an increase after three days of treatment compared to untreated (Appendix A). It was previously reported that the redistribution of MT1 to the nucleus is important for protection against oxidative and other cellular stresses [47,49,53,62]. The increase in nuclear MT1 that we observed in HIV-infected macrophages may be a response to increased oxidative stress induced by meth treatment as meth has been shown to increase ROS and oxidative stress in different cell types [63]. Increased nuclear MT1 occurred at different days throughout the time course suggesting that macrophages from different donors experienced cellular stress at varying timepoints with meth treatment, contributing to redistribution of MT1 to the nucleus.

### 2.5. Methamphetamine in the Absence or Presence of LPS Increases the Production of Inflammatory Mediators Associated with HIV-NCI and CNS Inflammation by HIV-Infected Macrophages

Cytokine production in macrophages and other cell types is regulated by NF-κB activation [64]. Interaction of nuclear MT1 with NF-κB is proposed to contribute to production of the cytokine TNF-α and IL-6 in macrophages [44]. Since we detected increases in MT1 nuclear localization, we examined whether this was associated with increased expression of cytokines and other inflammatory mediators by HIV-infected macrophages in response to meth treatment. For these experiments, monocytes from four independent donors were differentiated into macrophages, infected, and treated daily with meth or untreated for one to five days as described previously. In addition, separate cell cultures were treated with LPS during the last three hours of each day of untreated or daily meth treatment to characterize whether meth treatment in combination with an inflammatory stimulus would further increase inflammatory mediator production. We chose LPS because it was shown that MT1 is important for LPS induced IL-6 and TNF-α production in mouse macrophages [44], and because LPS is increased in the serum and CSF of PWH and thus is a biologically relevant inflammatory stimulus [65]. We first examined the kinetics of LPS-induced inflammatory mediator production by HIV-infected macrophages and found that 1 ng/mL for three hours were optimal for these studies. After each day of the five-day time course, cell supernatants were analyzed by Luminex multiplex assay for selected macrophage-related cytokines/chemokines and inflammatory mediators. The Luminex assay panel included mediators that have been shown to be correlated with HIV-NCI: IL-6, TNF-α, CXCL8 (IL-8), CXCL10 (IP-10), CCL2 (MCP-1), CCL8 (MCP-2), osteopontin (SPP1), and CX3CL1 (fractalkine). We also assayed for markers of inflammation elevated in PWH: CCL7 (MCP-3), CXCL12 (SDF-1), CCL5 (RANTES), and CXCL9 (MIG) [66]. The purpose of these experiments was to examine whether increased nuclear localization of MT1 is associated with enhanced production of these inflammatory mediators, in the presence or absence of LPS, in response to meth treatment.

We analyzed the data as fold change of inflammatory mediators produced with meth treatment compared to untreated, with untreated set to one, as well as fold change of mediators produced with meth + LPS compared to LPS alone, with LPS alone set to one, for each day of treatment. This enabled us to assess the impact of meth treatment on inflammatory mediator production, with and without LPS. There was large variation among donors in the fold change values for the mediators measured across the five days of treatment. Thus, we represented the data as maximal fold change for each mediator that occurred on any one of the five days of meth treatment for each donor.

Meth treatment in the absence of LPS resulted in a donor specific trend towards increased IL-6, TNF-α, CXCL10 (IP-10), CCL8, CCL7, and CXCL12, with cells from some donors exhibiting large fold increases in response to meth. Osteopontin was not changed overall with meth treatment, with cells from one out of four donors exhibiting increased expression (Figure 6A). IL-8 levels in untreated and meth treated samples were above the limit of quantification for two of the donors despite assaying many dilutions of the supernatants. Thus, we do not have a complete set of data for this cytokine and did not include IL-8 in Figure 6A. With LPS, there were significant increases in IL-8 and CCL8 with meth treatment, and a donor specific trend towards increased IL-6, TNF-α, CXCL10 (IP-10), osteopontin, CCL7, and CXCL12 (Figure 6B). For the most part, the largest fold increases in almost all of these inflammatory mediators in response to meth, both in the absence and presence of LPS, were detected in macrophages from the same donors. Fold increases of inflammatory mediators with meth treatment, in the absence and presence of LPS, for each donor at each day of treatment are shown in Appendix A. The amount of inflammatory mediators detected by Luminex assay for untreated, meth treated, LPS treated and LPS + meth treated macrophages for each donor at each day of treatment are shown in Appendix A. We found that HIV-infected macrophages produced very high amounts of CCL2 (MCP-1) which were above the limit of quantification for our assay at many dilutions. HIV-infected macrophages did not secrete detectable amounts of fractalkine and CCL5, and very low amounts of CXCL9.

These data indicate that meth treatment of HIV-infected macrophages increases LPS-induced IL-8 and CCL8. In addition, meth treatment, in the presence or absence of LPS, may be increasing the production of additional inflammatory mediators that have been implicated in the pathogenesis of HIV-NCI. We hypothesize that increased nuclear localization of MT1 in HIV-infected macrophages after chronic exposure to meth contributes to increased production of cytokines/chemokines and other inflammatory mediators, Therefore, this may be a mechanism by which chronic meth use, especially in the context of increased LPS levels in the CNS, may be exacerbating HIV neuropathogenesis. Future studies are necessary to characterize the direct effect of increased MT1 nuclear localization in macrophages on basal and/or LPS-induced cytokine/chemokine and inflammatory mediator production in the context of HIV infection and chronic meth use.

### 2.6. Donor Specific Increased ROS Levels in HIV-Infected Macrophages After Multiple Days of Methamphetamine Treatment

One of the major functions of MT1 is ROS neutralization [43,46,51]. Therefore, we assessed the association of increased MT1 gene expression and nuclear localization with ROS levels in HIV-infected macrophages after meth treatment. HIV-infected macrophages were treated with meth or untreated for one to five days as described previously and total ROS production was quantified by DCF fluorometric assay. We used pyocyanin, an ROS inducer [67] as a positive control, and hydrogen peroxide as a technical positive control for the assay and detected increased ROS production with both treatments compared to untreated. There were no significant changes in ROS levels with meth treatment compared to untreated for the six donors pooled at any timepoint assayed (Figure 7). However, ROS was at lower levels for most donors at early timepoints, while at later days of repeated meth treatment, ROS increased in a donor specific manner. It has been shown that meth increases ROS [42,68]. Our data suggest that chronic meth use may have an inhibitory effect on the anti-oxidant functions of MT1 that contribute to increased ROS in HIV-infected macrophages.

## 3. Discussion

Macrophages are important innate immune cells that perform a variety of functions critical to maintenance of tissue homeostasis. Some of these functions include phagocytosis, ROS production, antigen presentation, cytokine production, and wound healing, among many others [37,38,69]. The role of macrophages and their contributions to the pathogenesis of various inflammatory and autoimmune disorders has been the focus of many studies [70,71]. Maintenance of tissue homeostasis is a fine balance of many macrophage functions requiring complex signaling. Dysregulation of any of these functions may play a role in disease processes. The CNS contains immune cells, including macrophages and microglia [72,73], and they can contribute to the pathogenesis of inflammatory CNS diseases.

Peripheral blood monocytes that transmigrate across the BBB contribute to the CNS macrophage population, and monocytes in PWH that harbor HIV can enter the CNS, differentiate into infected macrophages, and establish and replenish the viral reservoir. Virus shed from these infected macrophages can infect microglia, and other resident macrophages, expanding the viral reservoir. Infected macrophages and microglia produce neurotoxic viral proteins including tat and nef, excess ROS, and cytokines, that can activate other parenchymal cells [9,10,11,13,14,15,16]. This contributes to a chronic neuroinflammatory environment that ultimately results in an accumulation of neuronal damage that manifests as HIV-NCI, even in the context of suppressive ART [17,18,19].

Meth use is a significant public health concern with use increasing in the general population in the United States. Data from the National Survey on Drug Use and Health from 2023 showed that 2.6 million people over the age of 12 used meth [26]. Meth is a potent CNS stimulant that produces feelings of euphoria and wakefulness and has been associated with cognitive impairment with long term use [74]. Meth use has also been shown to be associated with a greater risk of HIV acquisition [75]. There appears to be an increased prevalence of meth use among PWH [27,28]. Some studies have shown that meth use in PWH correlates with increased viral loads associated with ART non-adherence [32,33]. It was also shown that increased viral loads are correlated with meth use even in the presence of ART [34,35].

Macrophages are important mediators of HIV neuropathogenesis [10,11,17,40]. Thus, it is important to understand whether functional changes induced in macrophages by chronic meth exposure could exacerbate HIV-NCI. There are many studies on the impact of short-term meth exposure on macrophages, both with and without HIV [42,76,77,78,79], but there are limited studies that model long-term meth use. Given that people with HIV may use meth chronically for extended time periods, our experimental system of repeated daily meth treatment for five days enables us to characterize the effects of meth in that context. We demonstrated increased expression of MT1 gene isoforms in primary human HIV-infected macrophages with repeated meth treatment, which, to our knowledge, is the first report of MT1 expression in macrophages in the context of HIV and substance use. This meth treatment protocol did not increase MT1 gene isoforms in uninfected macrophages, indicating that meth-mediated increases in MT1 gene expression occur in the context of HIV infection.

MT1 protects cells from stress conditions including oxidative stress, cell death, metal toxicity, and DNA damage. MT1 is induced in a tissue and cell type specific manner by a variety of factors, including heavy metals, oxidative stress, glucocorticoids, and inflammatory factors, including LPS, IL-1β, TNF-α, and IL-6 [45,46,51,80]. When localized to the cytoplasm, MT1 can neutralize ROS and metals by directly scavenging these molecules [43,45,46,51]. During cellular stress, MT1 can also localize to the nucleus resulting in upregulation of its own transcription, and of oxidative defense mechanisms, and protection against apoptosis [43,52,53,80]. Some studies have also shown that MT1 contributes to cytokine production through modulation of NF-κB activity in some cell types [44,51]. Whether MT1 has inflammatory or anti-inflammatory activity appears to be context and cell type dependent, in part due to differences in the ability of MT1 to mediate NF-κB activation or inhibition [47,48,49,51,52,53]. One study showed that nuclear MT1 can be inflammatory in macrophages by contributing to cytokine production through facilitation of NF-κB activity [44]. It is postulated that MT1 can also serve as a zinc reservoir and its presence in the nucleus can modulate zinc levels that affect the activity of some transcription factors, including NF-κB [45,48,49,51].

Immune cell signaling and functioning are zinc dependent and metallothioneins are emerging as regulators of immune responses [51,81]. It has been shown that bacterial and viral infections, including influenza, upregulate MT1, and this upregulation was described as a cellular response to mitigate oxidative stress induced by the subsequent anti-bacterial or anti-viral inflammatory response [82,83]. There is some literature on MT1 expression or function in monocytes and macrophages [84,85,86,87]. A study demonstrated increased MT1 expression in HIV-infected monocytes from PWH during acute infection and after ART interruption that was described as a protective mechanism against apoptosis [88]. To our knowledge there are no studies about MT1 in the context of substance use disorder or in HIV-infected human macrophages. Thus, our current study was designed to examine MT1 in HIV-infected macrophages in the context of meth use.

To model chronic meth use in PWH, we infected macrophages with HIV and cell cultures were untreated or treated daily with meth for five days. RNA isolated from untreated and treated macrophages were then analyzed by RNA-seq to identify differential gene expression mediated by repeated meth treatment. Increased expression of multiple MT1 isoforms were identified and verified by qRT-PCR analyses of additional treated macrophage cultures, with donor-specific maximal expression of MT1 isoforms occurring at different days after repeated meth treatment. We hypothesize that multiple factors may be contributing to increased MT1 expression in HIV-infected macrophages. Both HIV infection and meth may be increasing the production of cytokines, including TNF-α, that have been shown to increase MT1 expression [51]. Oxidative stress induced by repeated meth treatment is also likely to contribute to increased MT1 expression in HIV-infected macrophages [80]. The binding of meth to one of its target receptors, trace amine associated receptor 1 (TAAR1) may also be inducing MT1 gene expression. TAAR1 has been identified as a target of meth and activation of this receptor results in increased cyclic adenosine monophosphate (cyclic AMP) levels as well as activation of protein kinase C (PKC) [89,90]. Cyclic AMP directly induces MT1 transcription [91] and PKC was shown to regulate metal transcription factor 1 (MTF-1) mediated activation of MT1 transcription [92]. Zinc also appears to be a potent inducer of MT1 in monocytes [86,87]. It was shown that circulating monocytes from PWH experiencing HIV viremia have higher intracellular zinc levels than circulating monocytes from ART suppressed PWH. That study linked increased intracellular zinc levels to apoptosis resistance in HIV-infected monocytes [88]. Characterization of mechanisms involved in meth-induced MT1 expression in HIV-infected macrophages, including the role of increased zinc levels, will be the focus of future studies.

Using immunofluorescence staining and confocal microscopy, we identified increased nuclear localization of MT1 in HIV-infected macrophages in response to meth treatment. Repeated increases in oxidative stress induced by daily meth treatment likely contributes to increased MT1 nuclear localization as oxidative stress was reported to be a major inducer of nuclear trafficking of MT1 [62]. Nuclear MT1 has been shown to contribute to LPS-induced TNF-α and IL-6 in macrophages by a mechanism involving increased NF-κB activity [44], thus establishing an inflammatory role for nuclear MT1 in macrophages. Therefore, we examined whether increased nuclear localization of MT1 induced by meth is associated with increased production, in the absence or presence of LPS, of macrophage-related inflammatory mediators involved in HIV neuropathogenesis and HIV-NCI. Recent meta-analysis pooled data from 29 studies involving PWH and identified correlations between inflammatory mediators in the CSF and CNS inflammation and HIV-NCI [66]. We examined the effects of meth treatment on the production of macrophage-related inflammatory mediators that were implicated in the pathogenesis of HIV-NCI or CNS inflammation in this meta-analysis. We found a trend towards increased basal expression in the absence of LPS of IL-6, TNF-α, CXCL10, CCL8, CCL7, and CXCL12 after repeated meth treatment of HIV-infected macrophages that was donor specific. With LPS, there was a significant increase in IL-8 and CCL8 with a trend toward increased IL-6, TNF-α, CXCL10, osteopontin, CCL7, and CXCL12 that was donor specific after meth treatment.

LPS-induced mediators significantly increased with meth treatment exhibit a variety of inflammatory functions. IL-8 is a cytokine that attracts neutrophils and monocytes [93] and increases the adhesion of monocytes to the vasculature [94]. It is important to note that supernatants from infected macrophages treated with meth in the absence of LPS produced high amounts of IL-8, above the limit of quantification in all dilutions tested. This suggests that HIV-infected macrophages chronically exposed to meth, may be a major source of IL-8 in the presence or absence of LPS. CCL8 exhibits a variety of inflammatory functions including chemoattraction of immune cells, including T cells and monocytes [95]. Our data also suggest that HIV-infected macrophages, in a donor-specific manner, produce more IL-6 and TNF-α with meth treatment in the absence or presence of LPS. Nuclear MT1 was reported to contribute to LPS-induced TNF-α and IL-6 in macrophages [44]. IL-6 is an inflammatory cytokine that promotes monocyte differentiation into macrophages, and pathogen clearance [96], and also facilitates lymphocyte trafficking and polarization [97]. TNF-α has been shown to upregulate adhesion molecules on endothelial cells [98], which is implicated in increased recruitment of immune cells into the brain, contributing to HIV-NCI [99,100]. Both IL-6 and TNF-α are neurotoxic [101] and are elevated in other neurodegenerative conditions including Alzheimer’s disease [102]. It was also found that PWH with HIV-NCI have higher levels of plasma IL-6 compared to PWH without HIV-NCI [103].

These data indicate that meth treatment of HIV-infected macrophages increases LPS-induced IL-8 and CCL8. In addition, meth treatment, in the presence or absence of LPS, may be increasing the production of additional inflammatory mediators that have been implicated in the pathogenesis of HIV-NCI. A limitation of our inflammatory mediator assays is that we did not specifically optimize our experimental design and treatment timings for each individual mediator. Thus, we may not have been optimally capturing production and secretion kinetics. Another limitation of this study is that we were unable to perform MT1 immunofluorescence and inflammatory mediator assays on macrophages from the same donor and are thus unable to make direct correlations between MT1 nuclear localization and inflammatory mediator production. We do not obtain enough monocytes from one leukopak to perform more than one assay with all timepoints. We therefore could not directly make correlations between our individual findings, especially given the donor-to-donor variability. Overall, our study suggests that increased nuclear localization of MT1 in HIV-infected macrophages after chronic exposure to meth contributes to increased production of LPS-induced IL-8 and CCL8, in addition to other cytokines/chemokines and inflammatory mediators. Increased CNS chemokines can recruit peripheral uninfected and HIV-infected monocytes into the CNS over extended periods of time, replenishing viral reservoirs and contributing further to ongoing chronic neuroinflammation, resulting in neurotoxicity that leads to neuronal damage and HIV-NCI [10,17,18,19]. Therefore, this may be a mechanism by which chronic meth use, especially in the context of increased LPS levels in the CNS, may be exacerbating HIV neuropathogenesis. Future studies are necessary to confirm the direct stimulatory effects of nuclear MT1 in macrophages on basal and/or LPS-induced inflammatory mediator production in the context of HIV infection and chronic meth use.

We also characterized the impact of repeated meth exposure on ROS levels because increased ROS production in the CNS is characteristic of HIV-NCI, and macrophages are major producers of ROS [104,105]. Studies have shown that meth induces ROS production in macrophages [42] and microglia [68,79]. ROS are highly reactive oxygen-containing species that are produced and have a role during normal cell processes such as metabolism and signaling but can be pathogenic and damaging to DNA, proteins, and lipids [106]. In the case of macrophages, ROS are also important mediators of inflammation and pathogen clearance [104,105]. Cells have many mechanisms to mitigate ROS by neutralizing this toxic species [107,108], including neutralization by MT1 [43,45,51,109].

Although we did not detect significant changes in ROS levels over the time course of meth treatment, ROS levels were at lower levels for most donors at early timepoints, while at later days of repeated meth treatment ROS increased in a donor specific manner. Our data suggest that repeated meth treatment may mediate ongoing oxidative stress, and MT1 and other ROS response mechanisms may be effective to neutralize it at early timepoints. However, with repeated exposure, those defense mechanisms may be unable to compensate, resulting in increased ROS levels. One limitation for these experiments is that we were unable to perform the MT1 immunofluorescence and ROS assays on macrophages from the same donor and are thus unable to make direct correlations between MT1 nuclear localization and ROS levels. Additionally, some studies showed that ROS is produced in a short time frame following meth treatment [77,78]. Future studies can include assaying for ROS immediately after the addition of meth for each day of the five-day time course.

While using primary MDM is a major strength of our study, it is also a limitation in that it yields variability among donors. Primary human cells from individuals respond differently; however, these responses more closely reflect what occurs in vivo. Additionally, the susceptibility of primary macrophages to HIV infection varies and although we did not detect significant differences in infection levels with meth treatment, a donor that is more highly infected may have a higher baseline level of cytokine production than a donor that is less infected. In macrophages from the five donors with which RNA-seq was performed, we did not find any trends in terms of higher as compared to lower infection levels and MT1 gene expression fold change values with meth treatment.

Another limitation of our studies is that we did not treat cells with antiretroviral drugs, and thus our findings represent that of uncontrolled viremia. The expression of *MT1X*, *MT1H*, *MT1G*, and *MT1E* was reported to be increased in monocytes from participants experiencing HIV viremia compared to uninfected individuals [88]. It was also reported that *MT1X* and *MT1G* are increased in the same individuals with acute viremia after treatment interruption compared to before treatment interruption [88]. It is possible that our findings on meth-mediated increases in MT1 gene expression and nuclear localization, and their association with cytokine and ROS production, would be different in HIV-infected macrophages treated with ART. This will be addressed in future studies.

Macrophages are heterogeneously infected and not every cell will have HIV DNA incorporated into their genome, termed HIV^+^, compared to cells that do not have HIV DNA and are only exposed to viral proteins, or HIV^exposed^. It is important to make this distinction because these two populations of cells may behave differently, as our laboratory has shown previously in monocytes [110,111]. In our immunofluorescence studies we were unable to stain for HIV p24 to identify HIV^+^ macrophages and characterize nuclear MT1 in HIV^+^ compared to HIV^exposed^ cells. Future experiments will focus on characterization of meth-mediated effects on MT1 expression and nuclear localization in HIV^+^ compared to HIV^exposed^ macrophages in the presence or absence of ART.

To our knowledge, this is the first report that that characterizes MT1 expression in primary human HIV-infected macrophages and its upregulation with repeated meth treatment. We showed that meth increases nuclear localization of MT1 that is associated with increased inflammatory mediator production, with a trend towards increased ROS levels. Upregulation and nuclear localization of MT1 by meth in HIV-infected macrophages may be, in part, a protective mechanism. However, the concomitant increase in inflammatory mediator production suggests that there are also negative impacts of increased MT1 nuclear localization. These functional changes indicate that MT1 may contribute to meth-mediated exacerbation of HIV-NCI pathogenesis, identifying mechanisms that can be examined further to characterize macrophage dysfunction in PWH who use meth.

## 4. Materials and Methods

### 4.1. Monocyte Derived Macrophage Cell Culture

Leukopaks were obtained from either the New York Blood Center (New York, NY, USA) or the Gulf Coast Regional Blood Center (Houston, TX, USA). PBMC were isolated by density gradient centrifugation using Ficoll-Paque Plus (Cytiva, Malboro, MA, USA). For qRT-PCR and cellular toxicity experiments, 40 million PBMC were plated in 60 mm dishes. For bulk RNA sequencing experiments, 40 million PBMC or 5 million negatively selected CD14^+^ monocytes from PBMC (Mojosort negative isolation kit, BioLegend, San Diego, CA, USA) were plated in 60 mm dishes. For immunofluorescence staining experiments, and cytokine and ROS assays, monocytes were isolated from PBMC by positive selection using CD14 magnetic beads (Miltenyi Biotech, Bergisch Gladbach, Germany). For immunofluorescence staining, 200,000 isolated monocytes were plated on coverslips, and for cytokine and ROS assays, 300,000 isolated monocytes were plated in each well of 24-well plates. PBMC and monocytes were then cultured for six days in macrophage media containing DMEM, 10% fetal bovine serum (Gibco, Waltham, MA, USA), 5% human serum (Sigma-Aldrich, St. Louis, MO, USA), 1% penicillin-streptomycin (Corning Corning, NY, USA), 1% GlutaMax (Gibco), 1% HEPES (Teknova, Hollister, CA, USA) and 10 ng/mL M-CSF (Peprotech, Cranbury, NJ, USA), with media changed after three days, to facilitate monocyte differentiation into macrophages.

### 4.2. HIV Infection and Methamphetamine Treatment of Macrophages

For all experiments, macrophages were derived from PBMC or monocytes after six days of culture with M-CSF. Macrophages were washed twice with HBSS and infected with 20 ng/mL HIV_ADA_ in media supplemented with M-CSF for 24 h. HIV_ADA_ is a monocyte/macrophage tropic laboratory-adapted strain originally isolated from a person with AIDS. After 24 h of infection, media were changed and cells were cultured in fresh media with M-CSF for two additional days. At the three-day post-infection timepoint, macrophages were then untreated or treated daily with 50 μM meth (Cayman Chemical, Ann Arbor, MI, USA) for an additional five days. The concentration of meth used in our studies is representative of intermediate blood and brain concentrations of meth after a binge dose. Binge doses of 260 mg–1 g were estimated to produce 17–80 µM meth levels in the blood [112] with increased concentrations of meth in brain compared to serum detected during the first few hours after ingestion, as quantified in rat brains [113].

### 4.3. Measurement of HIV Infection Levels

HIV-infected macrophages derived from PBMC in 60 mm dishes were untreated or treated daily with meth as described previously. Supernatants were collected after each day of treatment and analyzed for HIV p24 capsid protein to measure HIV infection levels using an ultrasensitive alphaLISA (Revvity, Waltham, MA, USA), according to the manufacturer’s protocols. Fluorescence was measured using the VarioskanLux plate reader (ThermoFisher Scientific, Waltham, MA, USA) and sample concentrations of HIV p24 in pg/mL quantified using a standard curve generated with recombinant HIV p24.

### 4.4. Measurement of Cell Toxicity

HIV-infected macrophages derived from PBMC in 60 mm dishes were untreated or treated daily with meth as previously described. Supernatants from cultures after each day of treatment were analyzed for LDH release as a measure of cell death using the CytoTox-ONE Homogeneous Membrane Integrity Assay (Promega, Madison, WI, USA) according to the manufacturer’s protocol. Briefly, cells were lysed and used as a maximal lysis control for LDH release. LDH in supernatant samples was analyzed as percent of maximal LDH release based on fluorescence values measured using the VarioskanLux plate reader (ThermoFisher Scientific, Waltham, MA, USA). HIV-infected untreated and meth treated samples were normalized to their own respective maximal lysis conditions.

### 4.5. RNA Sequencing of HIV-Infected Macrophages

Macrophages in 60 mm dishes, derived from negatively selected monocytes from three independent donors and from PBMC from an additional two independent donors, were infected with HIV and then untreated or treated daily with meth as described previously. For RNA extraction, cells were homogenized using Trizol Reagent (Invitrogen, Waltham, MA, USA) followed by addition of chloroform (Milipore, Burlington, MA, USA) and transfer to phase lock gel tubes (Quantabio, Beverly, MA, USA). The organic layer was collected, and isopropanol (Sigma-Aldrich, St. Louis, MO, USA) and sodium acetate were used to precipitate RNA, followed by washes with 75% ethanol. Pellets were dried, resuspended in RNase free H_2_O and quantified on a Nanodrop 2000 (Thermo Scientific, Waltham, MA, USA) to determine concentration, 260/280 values and 260/230 values. All samples had 260/280 values above 1.9 and 260/230 values above 2.0, with 0.2 μg of RNA from each donor from each condition sent to Novogene (South Plainfield, NJ, USA) for bulk RNA-seq analysis.

### 4.6. RNA-Seq Data Analysis

The raw data generated from sequencing were mapped to human reference genome GRCh38/hg38. The resulting raw count matrix was used to perform the differential expression analysis with DESeq2 implemented by Patek^®^ Flow^®^ software V12.7.0. DEG data were grouped by macrophage functions. The fold change cutoff for DEG of interest was set to 1.2 and higher, or −1.2 and lower. The sequencing data have been deposited in the NCBI GEO repository. Data were deposited from three independent donors (accession number GSE307856) and two independent donors (accession number GSE307858) separately. Both datasets are available using the accession number GSE307859.

### 4.7. qRT-PCR of MT1 DEG

PBMC from additional independent donors were plated in 60 mm dishes, differentiated into macrophages, infected with HIV, and untreated or treated daily with meth, followed by RNA extraction as described previously. Conversion of RNA into cDNA was performed using SuperScript IV VILO Master Mix kit with ezDNase (Invitrogen) per manufactures’ protocol. Briefly, 2 μg of RNA, ezDNAse, ezDNAse buffer, and water were added to PCR tubes and incubated at 37 °C for two minutes to allow digestion of residual genomic DNA. Then, SuperScript VILO IV and water were added to tubes. Samples were then converted to cDNA using an Eppendorf Vapoprotect thermocycler (Eppendorf, Hamburg, Germany) and stored at −20 °C. Quantitative PCR for MT1 gene isoforms *MT1E*, *MT1F*, *MT1G*, *MT1H*, and *MT1X* was performed as validation of the RNA-seq data using Taqman Gene Expression assay primers (Applied Biosystems, Waltham, MA, USA) with 18s used as a housekeeping gene. Data were quantified using the 2^−ΔΔCt^ method to compare meth treatment to untreated. For maximal expression analysis, RNA was extracted after each day of the five-day treatment course and converted into cDNA. Using the 2^−ΔΔCt^ method, the day of highest meth-mediated expression increase for each MT1 isoform for each donor was chosen to represent peak or maximal expression increase.

### 4.8. MT1 Immunofluorescence and Confocal Microscopy

Monocytes were positively isolated, plated on glass coverslips (Fisher Scientific, Waltham, MA, USA), differentiated into macrophages, infected with HIV, and untreated or treated daily with meth as previously described. Macrophages after each day of treatment were fixed with 4% PFA for 15 min at room temperature (RT), washed with 1X PBS three times, permeabilized with 0.1% Triton-X100 (Sigma-Aldrich, St. Louis, MO, USA) for four minutes at RT and washed twice with 1X PBS. Coverslips were incubated with blocking solution ((ddH_2_O, 0.5 M EDTA, pH 8.0, 45% gelatin from cold-water fish skin (Sigma-Aldrich, St. Louis, MO, USA), IgG Free Bovine Serum Albumin (Sigma-Aldrich, St. Louis, MO, USA), horse serum (Sigma-Aldrich, St. Louis, MO, USA), human serum (Corning, Corning, NY, USA)) for one to two hours at RT. Blocking solution was removed, primary antibodies diluted in blocking solution were added, and coverslips were incubated overnight at 4 °C. Anti-mouse monoclonal MT1 (Clone UC1MT, Abcam, Cambridge, United Kingdom) was used at 5 μg/mL (1:200) and mouse IgG1 (Invitrogen, Waltham, MA, USA) was used as a concentration matched isotype control. Coverslips were then washed five times with 1X PBS for five minutes. AlexaFluor 488 goat anti-mouse IgG (Invitrogen, Waltham, MA, USA) at 5 µg/mL (1:400) was used as the secondary antibody and was added along with AlexFluor 647 Phalloidin (Invitrogen, Waltham, MA, USA) at 1:400 in block buffer for a one hour incubation at RT. Coverslips were then washed five times with 1X PBS for five minutes and then mounted on glass microscope slides (Fisher Scientific, Waltham, MA, USA) using ProLong Diamond Antifade Mountant with DAPI (Invitrogen, Waltham, MA, USA). Cells were visualized and imaged using a Leica SP8 Confocal microscope (Leica Microsystems, Wetzlar, Germany). Z-series images were taken at 63× magnification to capture MT1 in the nucleus. Images were analyzed by a blinded observer using Volocity 7 image analysis software (Quorum Technologies, Sacramento, CA, USA). MT1 fluorescence within each nucleus (at least 200 cells per condition) was used to quantify nuclear MT1. IgG isotype-matched antibody was used as a negative control. Mean fluorescence intensity was calculated for each treatment, IgG background fluorescence was subtracted, and then fold change values were generated by dividing the mean fluorescence intensity of meth-treated cells by that of untreated cells, with untreated cell fluorescence set to one. Data shown as maximal increase corresponds to the highest fold change value on any day of the five-day treatment course for each donor.

### 4.9. Measurement of Inflammatory Mediators

Monocytes were positively isolated, plated in 24-well dishes, differentiated into macrophages, infected with HIV, and untreated or treated daily with meth as previously described. For each day of the five-day treatment course, either media or 1 ng/mL LPS (Sigma-Aldrich, St. Louis, MO, USA) was added to some wells of untreated and HIV-infected macrophages for the last three hours. Supernatants were then collected after each day of treatment and pooled from replicate wells, spun at 1200 rpm for five minutes to remove cells and debris, aliquoted, and stored at −80 °C. Inflammatory mediators were measured by Luminex multiplex assay. The MILIPLEX PLEXpedition Screening Panel, a configurable Luminex multiplex immunoassay kit, was designed by and purchased from MilliporeSigma (Burlington, MA, USA) to assay for the following analytes: IL-6, TNF-α, CXCL8 (IL-8), CXCL10 (IP-10), CCL2 (MCP-1), CCL7 (MCP-3), CCL8 (MCP-2), CXCL9 (MIG), CXCL12 (SDF-1), osteopontin (SPP1), CCL5 (RANTES), and CX3CL1 (fractalkine). Samples were run on a Luminex xMAP Intelliflex System at either 1:4 or 1:10 dilutions for HIV untreated or meth treated samples and 1:25 or 1:40 dilutions for untreated + LPS or meth + LPS treated samples. Data from Luminex assays were analyzed using Belysa Immunoassay Curve Fitting software version 1.2 (MilliporeSigma, Burlington, MA, USA). Data were analyzed as fold change of meth treatment compared to untreated, with untreated set to one, and meth + LPS compared to untreated + LPS, with untreated + LPS set to one. Data are shown as maximal increase representing the highest fold change value on any day of the five-day treatment course for each donor.

### 4.10. Measurement of Reactive Oxygen Species

Monocytes were positively isolated, differentiated into macrophages, infected with HIV, and untreated or treated daily with meth as previously described. After each day of treatment, ROS production was measured by incubating cells for 1 h with 10 μM CM-H_2_DCFDA (Invitrogen, Waltham, MA, USA), a cell-permeable fluorogenic probe for total intracellular ROS, diluted in HBSS containing MgCl_2_ and CaCl_2_ at RT. After washing with HBSS, cells were treated with either 200 μM Pyocyanin (Cayman Chemical, Ann Arbor, MI, USA), a ROS inducer as a biological positive control, or 0.03% hydrogen peroxide as a technical positive control, for one hour. Fluorescence was detected in a VarioskanLux plate reader (ThermoFisher Scientific, Waltham, MA, USA) using an excitation wavelength of 495 nm and an emission wavelength of 520 nm. Background or autofluorescence was quantified with a no-dye control and subtracted from all measured values. Data are shown as fold change of fluorescence of meth treatment compared to untreated, with untreated set to one, for each day of the five-day time course.

### 4.11. Statistical Analyses

Statistical analysis was performed using Graphpad Prism v10 (Boston, MA, USA). For comparisons of paired samples with and without meth, paired *t*-tests were used for normally distributed data and Wilcoxon matched-pairs signed rank tests were used for non-normally distributed data. For fold change values, one sample *t*-tests were used for normally distributed data and Wilcoxon signed rank tests were used for non-normally distributed data. Significance during the five-day time course was determined for meth-treated compared to untreated cells on each individual day. For maximal meth-induced increases, the highest absolute increase induced by meth regardless of the day of treatment was identified and significance was determined by comparing meth-treated to untreated values on that one day.

## Figures and Tables

**Figure 1 ijms-26-08875-f001:**
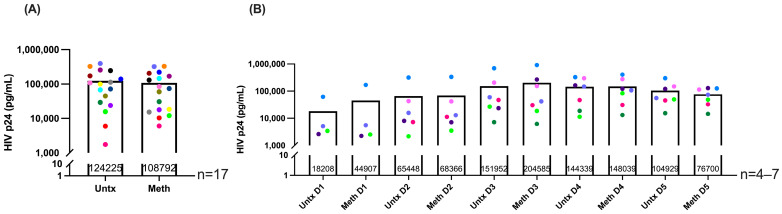
Methamphetamine treatment does not change HIV infection levels at any timepoint. HIV-infected macrophages derived from either PBMC, or from negatively or positively isolated monocytes were treated daily with methamphetamine or left untreated, and supernatants were collected. HIV p24 was measured in supernatants by ultrasensitive p24 alphaLISA and data are presented as amount of p24 in pg/mL. “Untx” refers to untreated cells. Colored points correspond to individual donors. (**A**) Represents supernatants from independent donors that were measured for HIV p24 levels after five days of meth treatment and (**B**) Represents supernatants from additional independent donors measured for HIV p24 levels after each day of treatment. Significance was determined by paired *t*-test for (**A**) (*p* = 0.2829). For (**B**), the Wilcoxon matched-pairs signed rank test was used for D1 (*p* = 0.8750), D2 (*p* = 0.5625), and D3 (*p* = 0.9375) and the paired *t*-test was used for D4 (*p* = 0.8535) and D5 (*p* = 0.3001).

**Figure 2 ijms-26-08875-f002:**
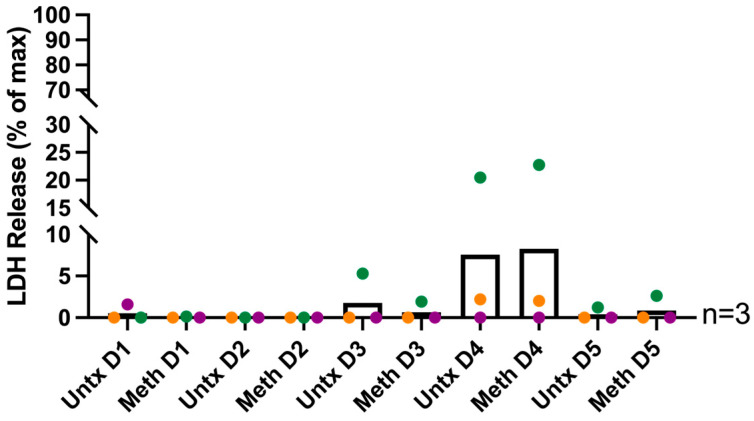
Methamphetamine treatment is not toxic to HIV-infected macrophages at any timepoint. HIV-infected macrophages derived from PBMC were treated daily with methamphetamine or left untreated, LDH assay was performed as a metric of cytotoxicity. Supernatants were collected and cells were lysed to serve as a positive control for 100% LDH release. Data are presented as a percent of maximum release for which the amount of LDH in the supernatants is divided by the LDH in the positive control and multiplied by 100. “Untx” refers to untreated cells. Colored points correspond to individual donors. No significance was determined by paired *t*-tests for untreated and meth treated samples for each day of treatment.

**Figure 3 ijms-26-08875-f003:**
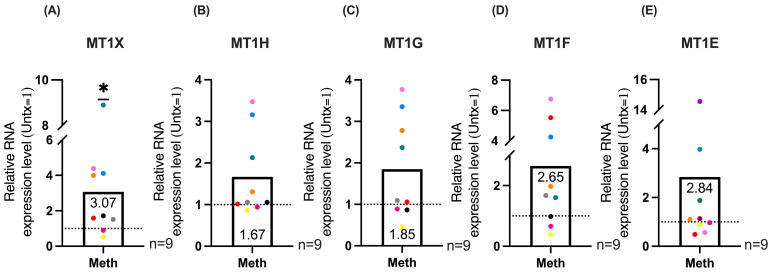
qRT-PCR of MT1 expression by HIV-infected primary human macrophages treated with methamphetamine. MT1 genes that were identified to be differentially expressed by RNA-seq were validated by qPCR. Primary human HIV-infected macrophages derived from PBMC were treated daily with methamphetamine for five days or left untreated. Graphs show relative mRNA expression levels of methamphetamine treatment compared to untreated, which is set to one, indicated by the dashed line. Bars represent mean. Colored points correspond to individual donors. “Untx” refers to untreated cells. (**A**) *MT1X* (**B**) *MT1H* (**C**) *MT1G* (**D**) *MT1F* (**E**) *MT1E*. Significance determined by Wilcoxon signed rank test for (**A**) (*p* = 0.0195) and (**E**) (*p* = 0.4258) and one sample *t*-test for (**B**) (*p* = 0.0634), (**C**) (*p* = 0.0723), and (**D**) (*p* = 0.0828). *, *p* < 0.05.

**Figure 4 ijms-26-08875-f004:**
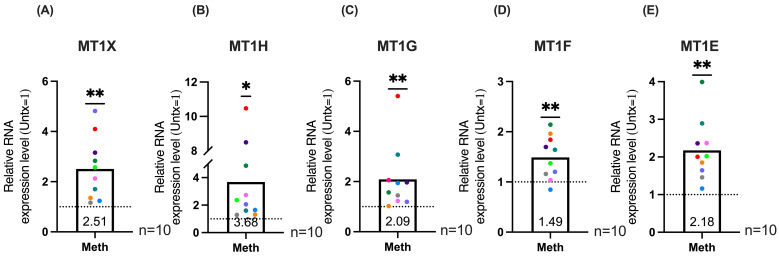
Maximal expression analysis demonstrates significant increases in MT1 gene expression. HIV-infected macrophages derived from PBMC were treated with methamphetamine or left untreated, and RNA was extracted, cDNA synthesized, and qPCR performed to quantify MT1 gene expression after each day of daily treatment. Maximal expression was determined by identifying the value of greatest increase in gene expression per donor per gene. Graphs show relative mRNA expression levels of methamphetamine treatment compared to untreated, which is set to one, indicated by the dashed line. Each point represents the highest gene expression increase per donor for each gene Bars represent mean. Colored points correspond to individual donors. “Untx” refers to untreated cells. (**A**) *MT1X* (**B**) *MT1H* (**C**) *MT1G* (**D**) *MT1F* (**E**) *MT1E*. Significance determined by one sample *t*-test for (**A**) (*p* = 0.0041), (**B**) (*p* = 0.0288), (**D**) (*p* = 0.0057) (**E**) (*p* = 0.0013), and by Wilcoxon signed rank test for (**C**) (*p* = 0.0020). *, *p* < 0.05, **, *p* < 0.01.

**Figure 5 ijms-26-08875-f005:**
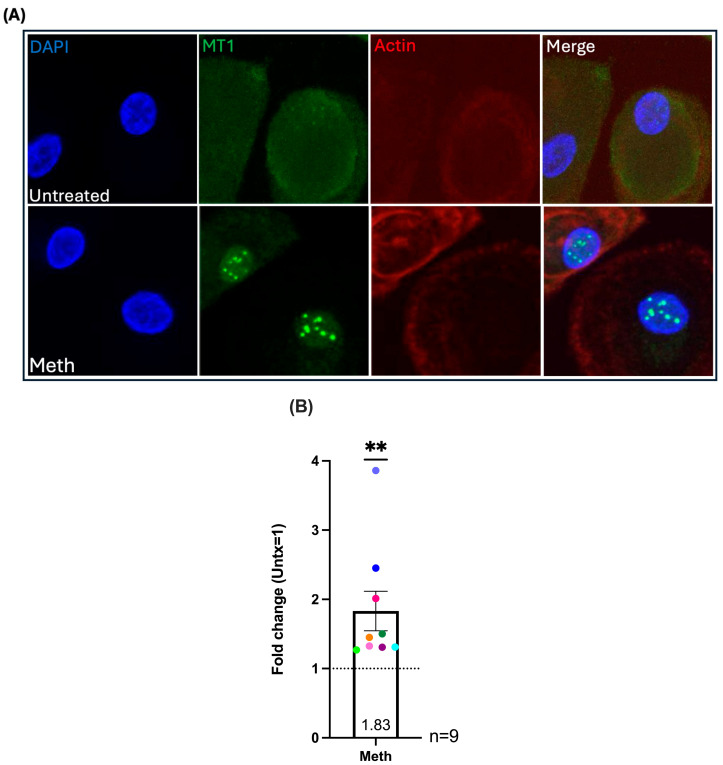
Methamphetamine treatment increases nuclear localization of MT1. HIV-infected macrophages derived from positively isolated monocytes were treated with methamphetamine or left untreated and fixed and stained for MT1 protein after each day of treatment. (**A**) Images from a representative donor with and without meth. Z-series images acquired at 63× to clearly observe nuclei and measure nuclear MT1. MT1 pixel intensity per nucleus was measured for at least 200 cells per condition using Volocity 7 image analysis software. (**B**) Maximal increase in nuclear MT1. Fold change values of mean pixel intensity of nuclear MT1 were generated for each day of meth treatment compared to untreated, set to one. The highest fold change value of nuclear MT1 per donor is pooled, representing maximal nuclear localization. Untreated is set to one, as indicated by the dashed line. Colored points correspond to individual donors. “Untx” refers to untreated cells. Significance determined by Wilcoxon signed rank test for (**B**) (*p* = 0.0039). **, *p* < 0.01.

**Figure 6 ijms-26-08875-f006:**
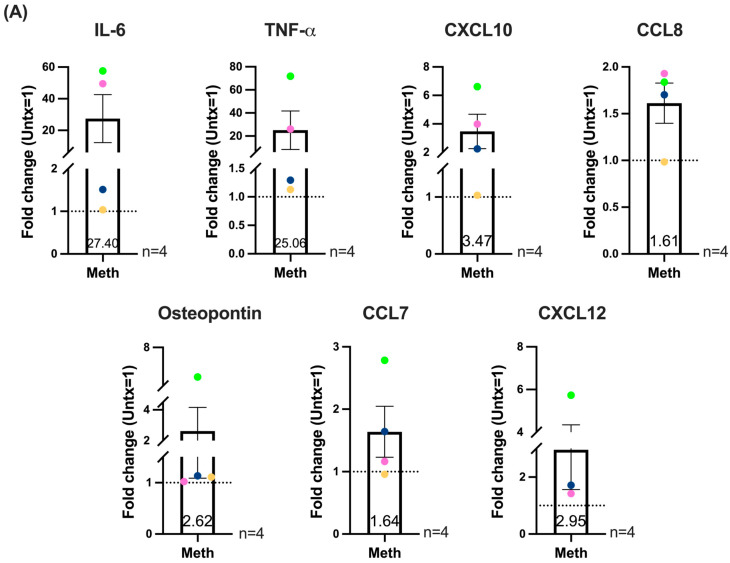
Methamphetamine with and without LPS treatment appears to increase inflammatory mediators shown to correlate with HIV-NCI or inflammation in PWH. HIV-infected macrophages derived from positively isolated monocytes were treated daily with methamphetamine or left untreated. In additional cell cultures, media or LPS was added at 1 ng/mL for the last three hours of meth treatment. This was done for each day of meth treatment. Supernatants were collected and pooled from replicate wells, aliquoted, and stored at −80 °C. Inflammatory mediator production was measured by Luminex multiplex assay. Fold change values for meth treatment compared to untreated, set to one, and for meth + LPS compared to LPS alone, set to one, were calculated. Data are presented as maximal fold change increase of meth treatment over untreated and fold change of meth + LPS over LPS for each mediator. Untreated or LPS is set to one, as indicated by the dashed line. Colored points correspond to individual donors. “Untx” refers to untreated cells. (**A**) Maximal increase of meth compared to untreated for each inflammatory mediator. Top row (left to right): IL-6, TNF-α, CXCL10, CCL8. Bottom row (left to right): Osteopontin, CCL7, CXCL12. (**B**) Maximal increase of meth + LPS compared to LPS for each inflammatory mediator. Top row (left to right): IL-6, TNF-α, IL-8, CXCL10. Bottom row (left to right): CCL8, osteopontin, CCL7, CXCL12. Significance determined by one sample *t*-test for the following mediators in (**A**) IL-6 (*p* = 0.1803), TNF-α (*p* = 0.2442), CXCL10 (*p* = 0.1343), CCL8 (*p* = 0.0650), CCL7 (*p* = 0.2951), CXCL12 (*p* = 0.2167). One sample *t*-test for the following mediators in (**B**) IL-6 (*p* = 0.1885), TNF-α (*p* = 0.1840), IL-8 (*p* = 0.0429), CXCL10 (*p* = 0.0605), CCL8 (*p* = 0.0229), osteopontin (*p* = 0.0699), CCL7 (*p* = 0.1633), and CXCL12 (*p* = 0.2247). Wilcoxon signed rank test for (**A**) osteopontin (*p* = 0.1250). *, *p* < 0.05.

**Figure 7 ijms-26-08875-f007:**
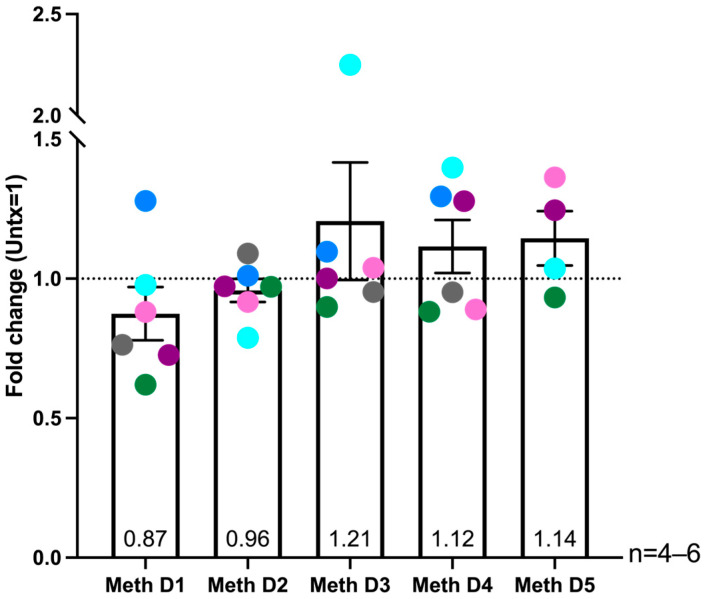
Methamphetamine treatment may result in increased ROS at in some donors. HIV-infected macrophages derived from positively isolated monocytes were treated with methamphetamine daily or left untreated and total reactive oxygen species (ROS) levels were measured. Cells were incubated with DCF dye (10 μM) diluted in HBSS + MgCl_2_ + CaCl_2_ for one hour, followed by incubation with 200 μM pyocyanin or 0.03% hydrogen peroxide for one hour as positive controls. Fluorescence was detected in a fluorimeter. Data were analyzed as fold change of treatment compared to untreated, with untreated set to one. Data are shown as fold change values for each day of meth treatment compared to untreated, as indicated by the dashed line at one. Colored points correspond to individual donors. “Untx” refers to untreated cells. Significance determined by one sample *t*-test for D1 (*p* = 0.2456), D2 (*p* = 0.3592), D3 (*p* = 0.3724), D4 (*p* = 0.2781), and D5 (*p* = 0.2354).

**Table 1 ijms-26-08875-t001:** Methamphetamine increases expression of MT1 genes in HIV-infected macrophages.

Gene	HIV + Meth vs. HIV Untreated				
	Fold Change				
	Donor 1	Donor 2	Donor 3	Donor 4	Donor 5
*MT1X*	0.472221266	2.333334	1.972971	1.28114	1.437185
*MT1H*	0.81538437	3.079997	2.800008	2.060447	1.731623
*MT1G*	0.330471042	8.647055	1.759038	1.879654	1.511675
*MT1F*	3.415092533	0.961539	1.284214	1.502437	1.587024
*MT1E*	0.747127113	7.874976	1.632653	1.072625	1.273819

## Data Availability

RNA-sequencing data are available on the NCBI GEO repository with the accession number GSE307859.

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
