# Peer review of "Methamphetamine Induces Metallothionein 1 Expression and an Inflammatory Phenotype in Primary Human HIV-Infected Macrophages"

_ijms, 2025, doi:10.3390/ijms26188875_

Round 1
Reviewer 1 Report
Comments and Suggestions for Authors
Weiselberg and colleague report a role of methamphetamine (METH) in regulating metallothionein 1 (MT1) expression in monocyte-derived macrophages (MDMs) infected with HIV-1. The authors find that METH increases expression of MT1 in MDMs and production of soluble factors associated with neuroinflammation and/or neuronal disorders including proinflammatory cytokines. Although this study does not include mechanistic insights into these phenotypes, findings in this manuscript might help us understand how METH exacerbates HIV-associated neurocognitive disorders. However, there are concerns to be addressed to improve the clarity and rigor of the study. Some revisions are suggested below.
Since uninfected MDMs were not examined, it’s not clear if findings in this study are driven solely by METH or there are additional/synergistic effects of METH on top of the phenotypes induced by HIV-1 infection. Data from uninfected MDMs are needed.
% of HIV infection in MDMs should be included in the manuscript in addition to p24 production data. Neurocognitive dysfunction is observed in PWH on suppressive therapy as well where the number of HIV+ macrophages is extremely low. Have you tested if a lower moi where only a few % or even less than 1% of MDMs are infected can induce MT1 expression and soluble factors with METH? Although the authors stated the system in this study was a model for uncontrolled infections, this point is important to consider.
Immunofluorescence images lack HIV-1 staining. The authors stated that they could not successfully stain p24Gag. However, it is important to show if MT1 upregulation and nuclear localization is seen only in HIV+ cells or bystander cells as well. If p24Gag staining does not work, suggest GFP-expressing HIV-1, for example.
Table 1 should be replaced with a graph showing values with a dot representing an individual donor like other graphs.
For time course experiments (Fig.1,2,8), 2-way ANOVA (if data are normally distributed) should be used instead of t-tests on each day. There are two factors.
Fig 5, MT1 expression seems to be localized in membrane blebbing in untreated cells. What are they? Are they representative?
Please include row values for Fig. 6&7, maybe in supplement. All data are normalized, and we cannot discuss if a 1.5-fold enhancement of CCL8 has a biological significance, for example. The enhanced values could still be just above the detection limit and/or a baseline. Once again, data from uninfected MDMs are needed.
Please explain/justify why 5-day METH treatment is considered a “long-term” treatment.
Discussion can be more concise. On the other hand, there is no discussion about the mechanisms underlying MT1 up-regulation and cytokine secretion with METH.
Discussion (p16), “Macrophages in the brain are mainly derived from peripheral blood monocytes that transmigrate into the CNS” needs to be rephrased. Border associated macrophages in the brain including perivascular macrophages seem to be derived from yolk-sac and replenish locally unless significantly damaged (PMID: 27135602).
Author Response
Please see attached letter.

Reviewer 2 Report
Comments and Suggestions for Authors
This submitted manuscript investigated the effects on Meth on metallothionein expression in HIV-infected human macrophages and provided interesting data. However, the whole manuscript has no focus and some statements were not clear such as “Meth treatment may result in decrease ROS”. Only the Figures 3, 4, 5 discussed about Metallothionein and other figures seems be away from the center hypothesis. There was no mechanism study. Discussion was too long and no focus. The description on statistical analysis was also not clar
Author Response
Please see attached letter

Reviewer 3 Report
Comments and Suggestions for Authors
This study demonstrates that methamphetamine (METH) treatment of macrophages increases the expression and nuclear localization of metallothionein 1 (MT1), which is associated with enhanced cytokine production. The authors further propose that MT1 contributes to HIV-associated neurocognitive impairment (HIV-NCI).
A significant limitation of the study is that all experiments were conducted in vitro using monocyte-derived macrophages (MDMs). This paper did not provide any evidence showing that METH exposure increases MT1 expression or nuclear translocation in brain-resident macrophages or microglia either in vivo or in vitro. This raises concerns about the physiological relevance of the findings to CNS pathology.
Moreover, the discussion of macrophage reservoirs in HIV infection would benefit from citing more recent literature. In addition to references 8 and 16, updated studies should be included to reflect current understanding of macrophage reservoirs, particularly within the CNS.
The cells in this study were treated with 50 μM METH. The authors state that this concentration reflects intermediate blood levels observed after a binge dose and falls within the range relevant to brain concentrations. However, it would strengthen the study to explicitly report the typical METH concentrations in both blood and brain following binge use, to justify the physiological relevance of the in vitro dosing.
Author Response
Please see attached letter

Round 2
Reviewer 1 Report
Comments and Suggestions for Authors
The authors responded well to the previous critiques and have clarified the manuscript. There is no further concern from this reviewer.
Author Response
Please see attached letter to reviewers.

Reviewer 2 Report
Comments and Suggestions for Authors
This is a revised manuscript. However, the reviewer is not satisfied with the responses. The reviewer cannot identify the point-to-point response. The manuscript has been revised extensively and was very hard to follow. A concise version would be easy for reading. The reviewer still has major concerns on the conclusion of this manuscript. The manuscript claimed that Meth can increase metallothionein expression in HIV-infected macrophages but without leading to macrophage toxic and increasing HIV infection. The meth can increase cytokine production without LPS. However, the roles of MT in Meth mediated cytokine production were not established here. The authors should present data whether MT1 inhibition or knockdown can mitigate Meth induced cytokine production in HIV-infected macrophages.
Author Response
Please see attached letter to reviewers

Reviewer 3 Report
Comments and Suggestions for Authors
My main concern has not been adequately addressed. The authors found that METH treatment increases metallothionein expression in HIV-infected human macrophages in vitro and then suggest this may contribute to HIV-associated neuropathogenesis. They acknowledge the difficulty of obtaining brain myeloid cells and argue that peripheral monocyte-derived macrophages (MDMs) can serve as a proxy for residual CNS myeloid cells. If this is the case, there should be at least some validation at the tissue level to support this claim. Otherwise, the discussion related to neuropathogenesis should be removed or significantly toned down.
Author Response
Please see attached letter to reviewers

Round 3
Reviewer 3 Report
Comments and Suggestions for Authors
Please remove "Potential contributions to HIV neuropathogenesis" from the title, as well as all related descriptions in the introduction and discussion sections
Author Response
Please see attached letter with response
